# The biomechanical origin of extreme wing allometry in hummingbirds

Dimitri A. Skandalis[1], Paolo S. Segre[1,7], Joseph W. Bahlman[1], Derrick J. E. Groom[2,8], Kenneth C. Welch Jr.[2], Christopher C. Witt[3], Jimmy A. McGuire[4,5], Robert Dudley[5], David Lentink[6] & Douglas L. Altshuler[1]

Flying animals of different masses vary widely in body proportions, but the functional implications of this variation are often unclear. We address this ambiguity by developing an integrative allometric approach, which we apply here to hummingbirds to examine how the physical environment, wing morphology and stroke kinematics have contributed to the evolution of their highly specialised flight. Surprisingly, hummingbirds maintain constant wing velocity despite an order of magnitude variation in body weight; increased weight is supported solely through disproportionate increases in wing area. Conversely, wing velocity increases with body weight within species, compensating for lower relative wing area in larger individuals. By comparing inter- and intraspecific allometries, we find that the extreme wing area allometry of hummingbirds is likely an adaptation to maintain constant burst flight capacity and induced power requirements with increasing weight. Selection for relatively large wings simultaneously maximises aerial performance and minimises flight costs, which are essential elements of humming bird life history.

---

[1] Department of Zoology, University of British Columbia, Vancouver, BC, Canada V6T 1Z4. [2] Department of Biology, University of Toronto, Scarborough, ON, Canada M1C 1A4. [3] Department of Biology and Museum of Southwestern Biology, University of New Mexico, Albuquerque, NM 87131, USA. [4] Museum of Vertebrate Zoology, University of California, Berkeley, CA 94720, USA. [5] Department of Integrative Biology, University of California, Berkeley, CA 94720, USA. [6] Department of Mechanical Engineering, Stanford University, Stanford, CA 94305, USA. [7] Hopkins Marine Station, Department of Biology, Stanford University, Stanford, CA 93950, USA. [8]Present address: Biology Department, University of Massachusetts Amherst, Amherst, MA 01003, USA. Correspondence and requests for materials should be addressed to D.L.A. (email: doug@zoology.ubc.ca)

Flight requires specialised morphology and physiology, and among the extant flying animals, hummingbirds exhibit some of the most extreme adaptations[1–4]. Hummingbirds sustain hovering, a highly energetically costly behaviour supported by numerous morphological and kinematic innovations[3, 5, 6]. Perhaps as ecologically fundamental, hummingbirds are highly aggressive, with frequent aerial competitions determined by aerial agility[7, 8] and possibly influenced by differences in body size[9]. An often overlooked feature of hummingbird morphology is an unusually large increase in wing area with increasing body weight ($W = M_Bg$) compared to other birds[10]. The exponent of the allometric relationship (equations of the form $Y = aW^b$) of hummingbird wing area to body weight has been estimated between 1.1 and 1.3, compared to about 0.7 across all other birds[10, 11]. This large exponent indicates that larger species have very large wings for their body weight, even though larger wings are predicted to be negatively associated with many aspects of aerial agility[11] and so could compromise flight performance.

Understanding the origin of this wing area allometry and how it influences flight performance has the potential to explain how hummingbirds have diversified into their specialised ecological niche, and explain the biomechanical evolution of flying animals more generally. The challenges of studying allometric variation are to place calculated exponents into a functional context and to link patterns among species to variation within species[12, 13]. Addressing these challenges allows us to assess the possible significance and origin of proposed allometries.

Allometries linked to flight performance do not evolve in isolation. The coevolution of suites of biomechanical traits dictates organismal performance, resulting in patterns such as the dependence of flight performance allometry on species elevation[14]. The functional evolution of any one trait, such as wing area, must therefore be considered alongside many correlated biomechanical traits. Previous work has especially focused on the evolution of flight performance in response to changes in elevation[5, 14, 15], but a general theory linking this variation to the proximate determinants of flight performance has not yet been developed. Moreover, because allometries are evolving traits, a general understanding of the evolution of flight performance must start at the variation observed among individuals and populations. A barrier to such studies is the daunting number of traits that can potentially be related to flight performance, making it difficult to choose a suite on which to build a complete framework. Simultaneously, the large number of traits might suggest that there are many potential evolutionary paths resulting in similar flight performance. An integrative perspective on this problem must be able to explain not just the presence or absence of an allometry, but also explain its magnitude. We approach this general problem by considering the mechanisms that contribute to the generation and cost of aerodynamic force in flight, and thus develop a framework to unify many aspects of hummingbird flight physiology.

All animals that use powered flight must generate time-averaged forces to support their body weight, which therefore represents the minimum level of selection. Flight forces in excess of body weight can then contribute to other flight behaviours, such as aerial displays and aggressive encounters. The dependence of aerodynamic forces on kinematic and morphological parameters is encapsulated by well-known scaling relationships. According to a blade element model (developed in 'Methods'), the time-averaged equation for vertical, weight-supporting aerodynamic force during hummingbird hovering is

$$\overline{F}_V = W = \frac{1}{2}\rho \overline{U}^2 S \overline{C}_V, \qquad (1)$$

following the Buckingham $\pi$ theorem, where the mean force $\overline{F}_V$ is the product of air density ($\rho$), representing the association between body mass and the physical environment a hummingbird has selected; stroke-averaged wing velocity [$\overline{U} = 4f\Phi R_2$, where $f$ is stroke frequency, $\Phi$ is stroke amplitude and $R_2$ is the wing length corrected for the spanwise chord width distribution[16]]; wing surface area ($S$); and a dimensionless stroke-averaged force coefficient ($\overline{C}_V$) that subsumes evolved differences in wing morphology such as wing twist and camber, and dimensionless postural changes such as angle of attack. The aerodynamic force equation has conventionally been used to derive isometric predictions of the right-hand side terms[11, 17, 18] against which empirical relationships are then compared. However, because in this approach only isometries are explained by theory, we lack functional context in the more common situation that animals violate the isometric model.

Here, we develop an integrative allometric framework from aerodynamics principles to resolve the functional consequences of allometric variation in hummingbirds. We consider the sum of the individual contributions to weight support of each component of Eq. (1), while considering common sources of bias in phylogenetic comparative models, such as measurement error and phylogenetic uncertainty[19]. We find that among species, increasing weight support is derived entirely from increasing wing area, but within species, increasing weight support is derived from increases in both wing area and velocity. We then examine how this allometric variation affects the cost of flight behaviours and limits maximum performance. Among species, burst performance and flight costs are constant, because wing velocity is size invariant. Within species, however, the reliance on wing velocity for weight support leads to increased relative flight costs and diminished relative burst performance with increasing body weight. This framework applies equally among and within hummingbird species, providing an evolutionary pathway from intraspecific patterning to interspecific allometries.

## Results

**Modelling framework and data collection.** A general allometric version of Eq. (1) can be written as (omitting constants)

$$\log_{10}\overline{F}_V = b_{\overline{F}_V} \cdot \log_{10}W = \left( b_\rho + 2b_{\overline{U}} + b_S + b_{\overline{C}_V} \right) \cdot \log_{10}W, \qquad (2)$$

where each slope $b$ refers to a variable in Eq. (1), according to its subscript (derivation presented in 'Methods'). This model, which we term force allometry, offers two useful insights. First, the allometric exponents of the right-hand side variables must sum to the allometric relationship of force and body weight, $b_F$. For a weight-supporting force, $b_F \equiv 1$ as required by Eq. (1), and the right-hand side exponents must sum to unity. We consider below the alternative case that other slopes are possible when considering forces generated during flight behaviours that require greater than body weight support, such as burst maximum performance. This summation requirement is a fundamental check of the derived exponents that applies to all flying animals, because if it is not met, then some relevant parameters could be missing or badly estimated, and we may not confidently make predictions about the biological relevance of the allometries. A second essential result from this model is that because only the sum of the exponents in Eq. (2) is constrained, we predict a continuum of physical, morphological and kinematic strategies that can conceivably support weight, and the allometric exponents reveal which strategies are actually employed.

We have assembled a large data set that includes measurements of all components of Eq. (2) in birds generating weight-

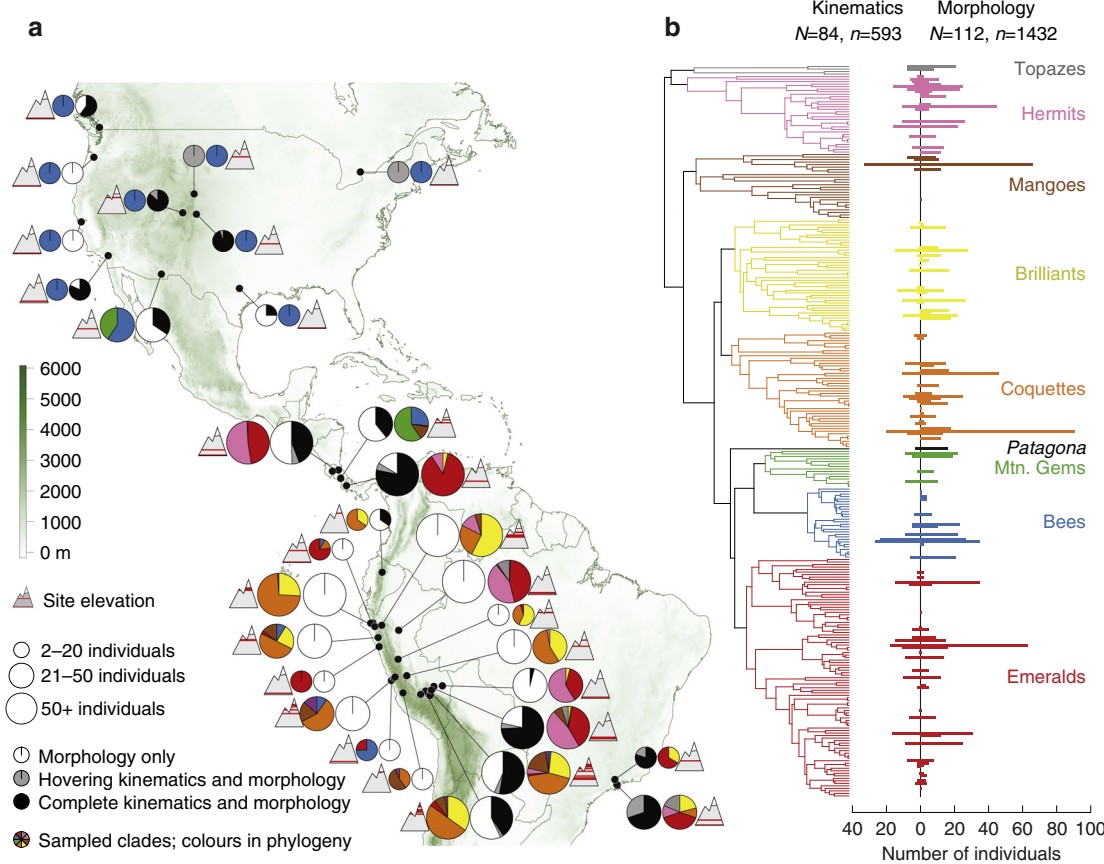

**Fig. 1** Biogeographic and phylogenetic sampling of hummingbirds. **a** Individual collection sites, grouping nearby sites in 5°×5° cells, along with the relative collection site elevation, biodiversity, and type of collected data (morphology, hovering kinematics, or hovering and load lifting kinematics). Colours in pie charts correspond to the colour scheme denoting humming bird clades in **b**. **b** All major clades of hummingbirds (defined by McGuire et al.[20]) were sampled both for kinematic and morphological parameters, though sampling effort varied widely across species and data type

supporting (hovering, $W$) and burst maximal (asymptotic load lifting, $\overline{F}_{burst}$) forces. These data collect up to 1500 individual records over 25 years (Supplementary Table 1), encompassing most of the biogeographic (Fig. 1a) and phylogenetic (Fig. 1b) distribution of the hummingbirds. Broad sources of uncertainty in the phylogenetic relationships among species in this study were visualised by ordination (principal coordinates, PC) and comparison to the species phylogeny published with McGuire et al.[20] (Fig. 2). The majority of variation among trees reflects uncertainty within the Hermit and Brilliant clades (PC 1, 33%), and further ambiguities within the Hermit clade alone (PC 2, 22%). All phylogenetic scenarios were sampled with equal probability, but the majority of trees fall into group i along with McGuire et al. (52% of trees), and only 6% of trees correspond to the largest topological differences from McGuire et al., group iv.

**Force allometry among and within species.** Our modelling procedures produce reliable inter- and intra-specific estimates of each allometry in Eq. (2), as judged by close agreement with the sum-to-one condition (weight support: $\Sigma b_{among} = 0.98$, $\Sigma b_{within} = 0.98$; Supplementary Fig. 1). Measurement error and phylogenetic relatedness impacted each variable differently even while maintaining the summation constraint (Supplementary Fig. 2). Phylogenetic uncertainty, as we model it here, altered mean exponents and credible interval (CI) widths by < 1%. Simulations in which we recalculate $\overline{C}_V$ under different conditions show that as long as measurement error is present in all variables, the summation condition is neither a trivial nor circular consequence of the calculation of $\overline{C}_V$ from the other components of Eq. (2)

(Supplementary Fig. 3, 'Methods'). Clade-wise examination of allometric exponents broadly confirms that the allometries we report are neither dominated by a single clade nor the result of averaging over many different clade-specific force-generating strategies (Supplementary Fig. 4).

Among hummingbird species, wing surface area scales almost exactly as one, $S \propto W^{1.01}$ (Bayesian CI: 0.908, 1.113; Figs. 3 and 4, Supplementary Table 1). In the context of the force equation, the sum-to-one rule predicts the other components are constrained to sum to zero, which is what we observe. Although it is possible that large hummingbirds could move to lower elevation, thus leading to a positive allometry with air density, there is no evidence that this occurs. Instead, we find a slight negative allometric exponent of air density ($\rho \propto W^{-0.06}$, CI: −0.112, −0.003) but this may depend on inclusion of outlier and poorly sampled species (Supplementary Fig. 5). Wing velocity among species is independent of body weight ($\overline{U} \propto W^{0.01}$, CI: −0.054, 0.074), in contrast to the isometric prediction that these should be positively correlated[11, 17] and derives from a constant stroke amplitude, coupled to a decline in stroke frequency proportional to the increase in wing length (Fig. 4). The force coefficient during weight support, $\overline{C}_{w,V}$, does not vary substantially ($\overline{C}_{w,V} \propto W^{0.01}$, CI: −0.122, 0.137), indicating that hummingbirds are dynamically similar in flight, unlike bats[18]. Among species, increasing weight support is therefore provided entirely by increasing wing area.

The reliance on increasing wing area to support body weight among species is not observed within hummingbird species (Figs. 3 and 4). Indeed, the average intraspecific pattern more

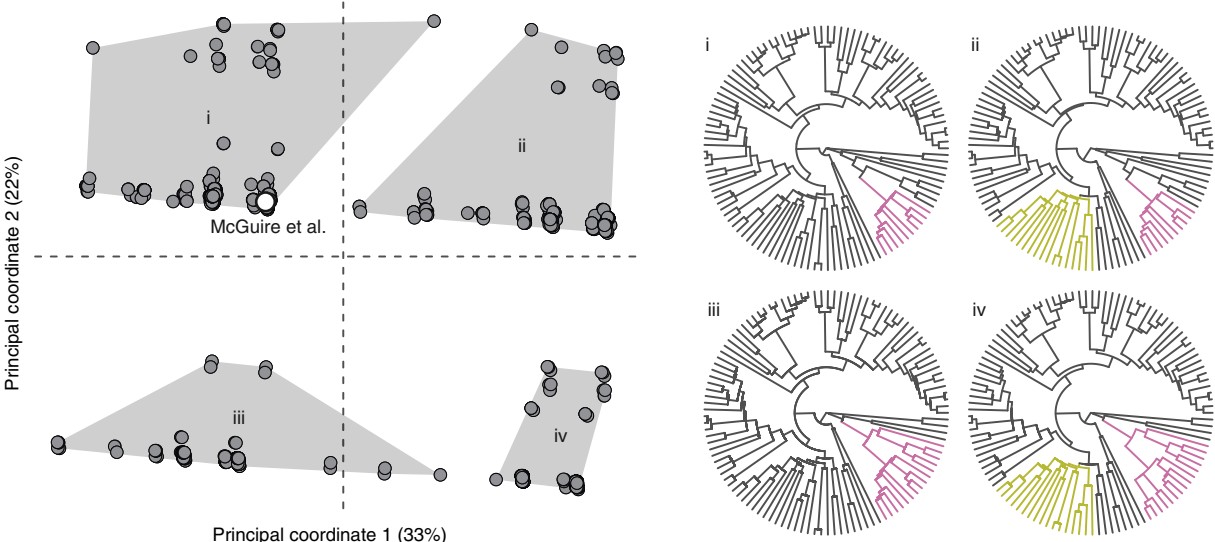

**Fig. 2** Uncertainty in phylogenetic relationships among species in this study. The variability in tree topology and branch length is mapped to a reduced-dimensional Euclidean space[35, 36]. The majority (55%) of uncertainty in species relationships is presented by two principal coordinates (PCs). Individual trees are shown by filled circles and clustered by similarity. To interpret the variability represented by the two PCs, we compare the median tree corresponding to each cluster (i–iv) to to the Maximum Clade Credibility (MCC) tree of McGuire et al.[20] The principal clade differences between the cluster median trees and the MCC are coloured in i–iv according to the scheme in Fig. 1. This method reveals that among species in this study (not hummingbirds overall), phylogenetic uncertainty primarily represents ambiguities in the Hermit and Brilliant clades. We allow for this uncertainty by integrating over many phylogenetic hypotheses

closely resembles biomechanical strategies suggested to occur among other bird species[11, 17]: weight support is provided by a combination of increasing wing area ($S \propto W^{0.42}$, CI: 0.366, 0.468) and wing velocity ($\overline{U} \propto W^{0.27}$, CI: 0.182, 0.354; Figs. 3 and 4). Intraspecific wing tip velocity increases with body weight due to constant stroke amplitude but unequal changes in stroke frequency and wing length (Fig. 4). Larger individuals tend to be associated with lower air densities at higher elevations, with an exponent similar to that found among species ($\rho \propto W^{-0.07}$, CI: −0. 0.085, −0.045; Fig. 4). A positive but uncertain change in $\overline{C}_{w,V}$ with body weight within species ($\overline{C}_{w,V} \propto W^{0.10}$, CI: −0.094, 0.289) must be interpreted cautiously until assigned to a specific cause, such as a systematic change in angle of attack.

**Allometry of burst flight performance.** We next examine the allometry of burst flight capacities through asymptotic load lifting, an unequivocal measure of maximum muscle capacity and performance that is predictive of manoeuvrability, foraging strategies and competitive ability[7, 8, 21]. This capacity can be expressed as the load factor, the maximum burst force as a proportion of body weight ($n = \overline{F}_{burst}/W$). Among species, load factor is size invariant ($n \propto W^{-0.01}$, CI: −0.112, 0.082; Figs. 3 and 4), indicating that manoeuvrability and competitive ability are independent of body weight. Conversely, within species load factor declines with body weight ($n \propto W^{-0.24}$, −0.364, −0.107), meaning that, on average, aerial performance is compromised in larger individuals. As for body weight support, we check the summation condition of Eq. (2) for burst performance, and find close agreement between the exponent of load factor and the sum of individual allometric exponents obtained during load lifting (Supplementary Fig. 1).

A key difference among and within hummingbird species is the extent of dependence on increasing wing velocity for increasing weight support, which can influence the energetic demands of flight. Wing velocity is a key determinant of specific induced power ($P^*_{ind} = P_{ind}/W$), which is the minimum power required to support weight[22, 23]. The scaling of profile and inertial powers are evaluated in greater detail in Supplementary Discussion, but our conclusions are unaffected by their inclusion.

Induced power is a function of the induced velocity, $\overline{v}_{ind}$, of the wake and of the wing velocity such that[22],

$$P_{ind} = nW \cdot \overline{v}_{ind} = nW \cdot \lambda \overline{U}_{wing}, \tag{3}$$

where $\lambda \equiv \overline{v}_{ind}/\overline{U}_{wing}$ is the dimensionless inflow ratio from actuator disc theory relating the mean wing velocity to the induced flow[22] and, like the force coefficient, depends on both wing morphology and kinematics. We again equate terms with body weight to develop an allometric expression for the scaling of specific induced power,

$$\log_{10}P^*_{ind} = b_{P^*} \cdot \log_{10}W = (b_n + b_\lambda + b_{\overline{U}}) \cdot \log_{10}W. \tag{4}$$

Equation (4) principally relates changes in specific induced power, load factor, and wing velocity. We cannot directly assess the contribution of inflow ratio, $b_\lambda$, because we have not measured the induced velocity, $\overline{v}_{ind}$, but we do not expect large differences among individuals and species with similar morphology and kinematics. Unlike the allometry of force in Eq. (2), the allometry of specific induced power, $b_{P^*}$, will vary depending on flight behaviour. For example, during hovering, the allometry of load factor is 0, and the allometry of specific induced power varies as a positive function of the allometry of wing velocity. In contrast, during maximum performance, the allometry of specific induced power is fixed at the maximum muscle capacity, and thus the allometry of load factor is a negative function of the allometry of wing velocity.

Specific induced power for hovering is constant among species (Figs. 3 and 4; $P^*_{w,ind} \propto W^{0.02}$, CI: −0.033, 0.063), but increases within species ($P^*_{w,ind} \propto W^{0.25}$; 0.193, 0.315). Burst specific induced power expended during load lifting, reflective of maximum muscle capacities, is independent of body weight both among ($P^*_{b,ind} \propto W^{0.07}$, CI: −0.21, 0.15) and within

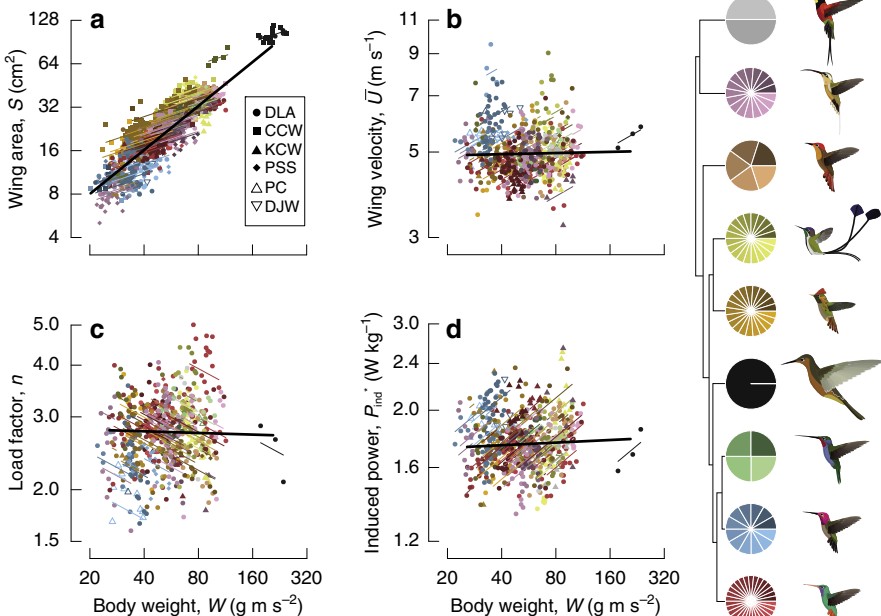

**Fig. 3** Allometric divergence among and within species. We contrast the slopes of wing area (**a**), wing velocity (**b**), load factor (**c**) and induced power requirements (**d**). The slope of each variable on body weight among species is shown in black, and each was calculated allowing for phylogenetic non-independence and measurement error. Individual records are shown along with the mean within-species slope fit through the respective empirical species means. Symbols denote collector. Individual observations and within-species slopes are coloured and shaded by species within clade, according to the cartoon phylogeny at right (colours as in Fig. 1). Sample sizes are provided in Supplementary Table 1

$(P_{b,\text{ind}}^* \propto W^{-0.03}$, CI: −0.212, 0.147) species. Reserve power, the difference in the allometries of maximum and hovering power, therefore declines in large individuals, but not large species. Overall, there is a decline in the production of burst vertical force relative to expended power in larger individuals, and although larger individuals proportionately expend the same maximum power during burst performance, they produce less relative force.

## Discussion

In principle, hummingbirds could adopt any one of multiple strategies to support changes in body weight during flight (Eq. (2)), expressed as movement to lower elevations with higher air density ($\rho$), increase in wing area ($S$), increase in wing velocity ($\overline{U}$) or adaptation of wing morphology and kinematics ($\overline{C_V}$). The potential contribution of each strategy differs; for instance, an order of magnitude in air density to support an order of magnitude in body weight is not possible. Each strategy may also entail tradeoffs, such as sacrificing potential habitats (air density allometry) or reconfiguring the wing (force coefficient allometry). We find that the allometry of force production among and within hummingbird species is solely a function of changes in the allometries of wing area and wing velocity. Among species, increasing weight support is provided exclusively by increasing wing area and maintaining constant wing velocity, whereas within species, weight support is provided both by increasing wing area and velocity. The advantage of maintaining constant wing velocity is apparent from Eq. (4), which shows that when $b_{\overline{U}} = 0$, expended power is only a function of the load factor, or reciprocally, the maximum load factor is only a function of the maximum available muscle power. The dependence on positive wing velocity allometry within species thus results in degrading burst force capabilities and escalating cost of flight in larger individuals. The extreme wing area allometry among hummingbird species appears to be an evolutionary strategy to mitigate the

performance and energetic disadvantages that would arise if the body plan of large species was extrapolated from intraspecific patterns.

The emergence of this extreme allometry among hummingbirds is likely due to pressures of their energetically demanding hovering flight and territoriality, frequently engaging conspecifics and confamilials in aerial bouts[9, 21]. Selection can therefore be expected to favour constant or minimally-increasing routine flight costs and burst aerial performance, which is supported by the weight independence of specific daily energy expenditure (DEE*) among hummingbird species[24], DEE* $\propto W^{-0.03}$. As observed, the force allometric pattern within species cannot be scaled up across the size range of hummingbirds without incurring severe penalties to both flight costs and burst forces. Maintaining burst performance margins could entail adaptation of the flight musculature, as may occur in other flying animals[25, 26] but the invariance of maximum available power among and within species suggests that hummingbirds' specialised muscles[4, 24] have reached the physiological limits of performance. Hummingbirds must therefore reduce energetic demand rather than supply, and increasing relative wing area is the simplest solution that both minimises flight costs and maximises performance.

Force allometry is a flexible method for examining the functional context of allometric variation in wing area. The approach can be applied among and within species to gain insight into the energetic and performance consequences of divergent force generation strategies. Separating the problem into its constituent components (Eqs. (2) and (4)) and then comparing the resulting exponents provides a framework for evaluating both the functional and statistical relevance of hypothesised allometries. This linear separation allows disparate data sets to be merged to provide consistent inference. Perhaps the most important insight from our framework is a shift in emphasis from single exponents intended to explain variation across all clades, to a nuanced view of possibly clade-specific balancing of weight-supporting strategies, including the possible contributions of the force

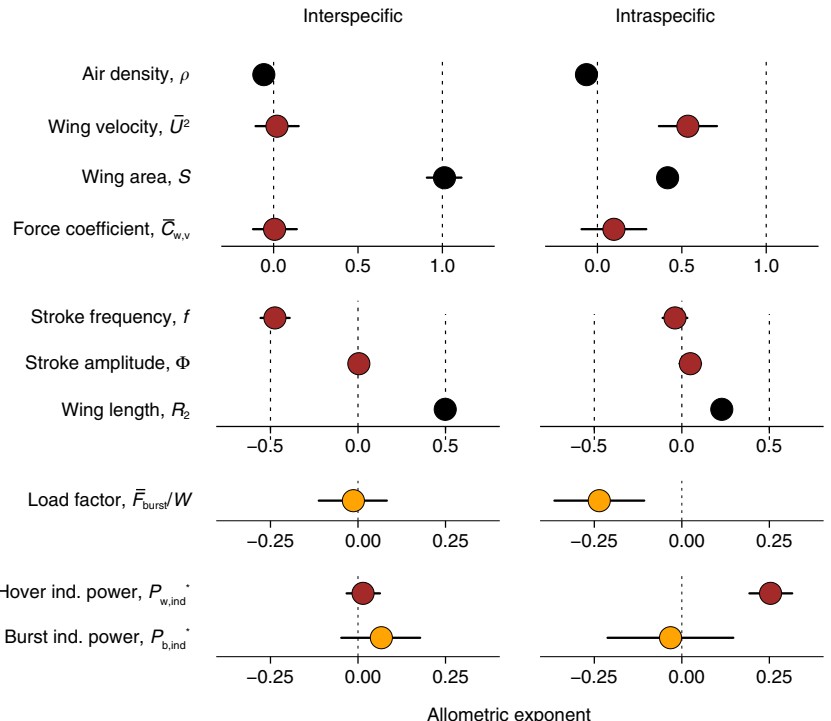

**Fig. 4** Comparison of allometric variation among and within species. Constant allometry of wing velocity among species coincides with constant burst force generation (load factor) and induced power. Positive allometry of wing velocity within species coincides with reduced load factor and escalating power requirements. The mean and 95% equal-tailed credible intervals of the posterior distribution of the allometric exponents are shown for each variable. Black circles are static morphological and environmental measurements, red circles were measured during hovering, and gold circles were measured during burst performance. Sample sizes are provided in Supplementary Table 1

coefficient[18]. We therefore applied our method to probe whether there is any evidence of variation in strategies among hummingbird clades. We find that the Bee clade has a uniquely low wing area exponent, comparable to that observed within species (Supplementary Fig. 4). This is particularly striking in light of the fact that the Bee clade is the most recently derived and most rapidly diversifying group of hummingbirds[20]. Combined with the observation that they also have uniquely low variation in body mass, this suggests a hypothesis that physiological diversification in the Bee clade is lagging behind species diversification. We have derived the equations here specifically for hummingbirds, but the force allometry approach can be applied to other flying animals with adjustments to account for the complexities of different wing strokes. This method could prove especially useful for quantifying subtle allometries in other families of flying animals, which likely operate in distinct selective regimes.

## Methods

**Data collection**. We use our allometric framework to analyse a data set obtained from individual hummingbirds sampled at different sites in Brazil, Canada, Costa Rica, Ecuador, Peru and the United States (Fig. 1). We do not explicitly distinguish sexes. Some kinematic and morphological data for these species have appeared elsewhere[7, 14, 27–30]. Sample sizes in each bivariate regression in numbers of species and individuals are presented in Supplementary Table 1, and decisions on species placement and taxonomy are provided in Supplementary Note 1. All data collection was performed in compliance with respective institutional guidelines. No randomisation or blinding was performed in this study. For the results reported here, we used all available samples, but investigated the impact of data subsets, as described below.

Air density was calculated from elevation using standard pressure and temperature relationships with elevation. We emphasise that in the context of this analysis, the allometry of $\rho$ is interpreted as evidence for an association between body mass and air density (or elevation), whether due to individuals or species selecting their environment or adapting to it, and not as hummingbirds effecting changes in local air density. Given a species' or individual's body mass, this regression is a prediction of the environment in which it will be found. In preliminary analyses, we found that $b_\rho$ was somewhat influenced by the inclusion of the unusually large and phylogenetically

distinct species *Patagona gigas*, and by inclusion of species with a single observation (Supplementary Fig. 5). Removal of these progressively reduces the air density allometric exponent toward zero, and so the overall influence of elevation and air density on species body mass is uncertain. Nonetheless, it is notable that the exponent is similar among and within species, which could indicate a common underlying mechanism. We investigated whether independent data sets might show evidence of a correlation between body mass and elevation. We collected species mean body masses and elevational midpoints from the Handbook of the Birds of the World (HBW[31]) and calculated mean species elevations from range maps provided by BirdLife International (BL[32]; see 'Methods' for further details of mapping procedures). Mean elevations from the two sources are well correlated (Supplementary Fig. 6), though with somewhat more error for low elevation species. Predictions of species maximum elevation were uncorrelated, likely because the range maps coarsely include all elevations within a contour. Elevation and body mass were examined using a phylogenetic regression implemented in MCMCglmm (see below). For all elevational parameters (minimum, mean, and maximum) in both data sets, the CIs of the slopes overlap 0 (Supplementary Fig. 6).

We examined whether capturing individuals at discrete sites influences results, because discrete sampling might not reflect continuous elevational distributions. We therefore sought to compare our results to independent estimations of species elevations, derived from species range maps[32]. Our observational data are reasonably well correlated with the derived species mean elevation and the distribution of species elevations (Supplementary Fig. 7).

Wing morphological variables were digitised from photographs of the spread wing as described by Altshuler et al.[14] or from wings spread on graph paper and traced in Adobe Illustrator (CCW collection). We obtained the wing area, $S$, and length, $R$, and second and third moments of area, $\hat{r}_2$ and $\hat{r}_3$, from these photos, and the aspect ratio was calculated as $AR = 4R^2/S$. There was a high degree of correlation in wing morphology and air density measurements among authors with overlapping species measurements (DLA, CCW and PSS data sets; Supplementary Fig. 8), and so apparent differences between data sets appear to be attributable to species sampling.

Kinematics (mean stroke amplitude and frequency) were digitised as previously described[14, 29, 30]. The mean wing velocity at the second moment of area was calculated as the product of stroke frequency, stroke amplitude, wing length, and the second moment of area, $(\bar{U} = 4f\Phi\hat{r}_2 R = 4f\Phi R_2$, see 'Methods'). Our results do not differ depending on this definition of wing velocity, or the use of the wing tip velocity directly, because $\hat{r}_2$ is not correlated with body mass (Supplementary Table 1). We calculated the vertical force coefficients in flight while hummingbirds support weight ($\bar{C}_{w,V}$) or during burst load lifting ($\bar{C}_{b,V}$), by rearranging Eq. (1).

**Phylogenetic uncertainty**. We allow for uncertainty in the phylogenetic hypothesis by integrating over a large number of phylogenetic scenarios. Suitable species phylogenetic hypotheses were derived from the posterior distribution of trees previously generated by BEAST analysis[20]. The tree posterior distribution comprised four chains run for one thousand generations each with a thinning rate of four, which we subsampled by half due to constraints on computer memory and run time. Inspection of the tree convergence suggested a burn-in period of 25 samples in the posterior was sufficient, yielding 450 trees (Σ). We then replicated these trees four times each in a procedure to account for uncertainties in species relationships created by different choices of individuals as species representative (this reflects a 1:4 ratio qualitatively balanced uncertainty and tree redundancy). In each replicated tree, for species in the phylogeny in which more than one individual was sampled, we randomly chose one individual as the species representative for that tree. The phylogenetic signal in the independent and dependent variables was allowed to be weaker than strict Brownian motion through Pagel's λ implemented as $\Sigma_\lambda = \lambda\Sigma + (1-\lambda)\mathbf{I}$, where $\mathbf{I}$ is the identity matrix[33, 34]. Phylogenetic independence and dependence are implied by λ = 0 or 1, respectively, and as we have no expectation for the phylogenetic strength, we assume a uniform distribution in this range[33].

We examine differences among the hypotheses represented in the posterior tree distribution using the method of Kendall and Colijn[35, 36]. Each tree is encoded by a score that reflects the extent to which the tree is completely described by the lengths or branching pattern of its edges. The set of scores then forms a Euclidean metric space, i.e. the difference between a pair of trees can be found by the difference in their scores. We visualise the broad uncertainty in the phylogenetic hypothesis by projecting the trees' pairwise distances into two principal coordinates[35], clustering of which revealed four subgroups of trees. Assuming each subgroup encapsulates a distinct source of phylogenetic uncertainty, we can summarise this uncertainty by finding the tree that lies at the geometric median of that subgroup, and then comparing this median tree to the Maximum Clade Credibility species phylogeny of McGuire et al.[20] Major topological differences are highlighted in Fig. 2. Because trees were pruned to the species available in this study, these results do not reflect overall sources of uncertainty in the phylogenetic hypothesis across all hummingbirds.

**Regressions and hierarchical bayesian modelling**. We used Markov Chain Monte Carlo (MCMC) simulations to analyse log–linear relationships[33, 37, 38]. For analyses presented in Figs. 3 and 4 and Supplementary Table 1, we model relationships while allowing for uncertainty in both the true, unobserved species means and in the phylogenetic hypothesis. We assumed flat, uninformative priors for the regression intercepts and slopes (normal distribution centred on zero with low precision, $\tau = 10^{-6}$), and for all standard deviations, σ (uniform distribution[39], on the interval 0–1000). Note that $\tau = \sigma^{-2}$, is the reciprocal of the variance. We model the unknown species means given potential intraspecific covariance by modifying the method of de Villemereuil[33] to include a minimally informative inverse-Wishart prior on the within-species covariance matrix. An alternative approach to within-species covariance in bivariate relationships is to place priors directly on the elements of the correlation matrix, but we found this led to poor mixing and a tendency to fixate on a correlation coefficient of $r = \pm 1$. The impact of modelling assumptions is compared in Supplementary Fig. 2.

We additionally examined whether there is any evidence that specific clades depart from the overall trends across all hummingbirds. The previous models, allowing for measurement error but not phylogenetic uncertainty, and with the previous uninformative priors, resulted in very wide CIs in some clades due to the reduced sample sizes. Because our objective was to find evidence for departures from the overall trend, we therefore used more reasonably informative priors. Following the overall trends, we employed a normal distribution with $\tau = 1$ and either a mean of 1, for wing area, or 0, for other variables. Other precisions were modelled directly through a weakly informative conjugate gamma prior with shape and scale equal to $10^{-3}$.

For each regression, we ran four parallel MCMC chains for ten thousand iterations each. The first five thousand samples of each chain were discarded as burn-in, yielding twenty thousand samples from the posterior. Whether a given slope credibly excluded a relevant value, especially zero, was assessed by comparing the overlap of the 95% equal-tailed CIs of the regression parameters to the reference value. We verified the trends reported here using the R package MCMCglmm[37] (uniform prior: V = 0, nu = 0; 25,000 iterations, 15,000 burn-in samples, three chains), including testing the effect of data subsets on the resulting exponents, especially the air density exponent (Supplementary Fig. 5). MCMCglmm did not support estimation of the unobserved species means, so intraspecific trends were calculated using the within-species centring method[40, 41].

**Force equation for flapping flight**. Dimensional analysis yields the familiar expression for steady aerodynamic force, $F = \frac{1}{2}\rho U^2 S C_F$ (noting that the force coefficient for hovering flight additionally absorbs differences in angle of attack). Because flapping wings generate unsteady forces, any allometric relationship for flight must consider a more general time-averaged approach to the vertical force

$$\overline{F}_V = \overline{\frac{1}{2}\rho U^2 S C_V},$$

where the velocity is calculated at the radius of gyration (second moment of area[16]). From this departure point, we can tailor the force equation to a form appropriate for the organisms of interest, by considering how the parameters vary over a stroke. In hovering hummingbirds, it is reasonable to assume that (i) wing area is constant through the stroke[42], (ii) air density is constant through the stroke and (iii) $\overline{U^2 C_V} = \overline{U^2} \cdot \overline{C_V}$, because by definition

$$\overline{C}_V = \frac{\overline{\frac{1}{2}\rho U^2 S}}{\overline{F}_V}.$$

Assumption (i) of constant wing area is not true for all flying animals, and we therefore derive the following equation specifically for hummingbirds,

$$\overline{F}_V = \frac{1}{2}\rho \overline{U}^2 S \overline{C}_V.$$

For convenience, we calculate the square of the average wing velocity, but for sinusoidal flapping motions, this differs from the average squared velocity only by a constant. We consider the instantaneous velocity of a flapping wing in hovering flight (body velocity = 0)[43] which is to within a good approximation a cosine function (zero velocity at tip reversal and maximal at midstroke)[42, 44],

$$U(t) = R_2\dot{\varphi}(t) = R_2\Phi 2\pi f\cos(2\pi ft),$$

where Φ and f are the mean stroke amplitude and frequency. Because the radius of gyration ($R_2$) can be assumed constant in hovering hummingbirds (but not for birds in general, for bats, or hummingbirds in forward flight), it is sufficient to calculate the average angular velocity (which is always positive),

$$\overline{\Omega} = \frac{1}{T}\int_0^T |\dot{\varphi}|\mathrm{d}t = \frac{1}{T}\int_0^T (\Phi 2\pi f) \cdot |\cos(2\pi ft)|\mathrm{d}t = 4\Phi f,$$

and therefore,

$$\overline{U} = 4\Phi f R_2.$$

Note that Φ here refers to the amplitude of the cosine function, one-half of the pronation-to-supination amplitude used elsewhere. Substituting this difference in definition, $\overline{U} = 4(\Phi/2)fR_2 = 2\Phi fR_2$ as in Ellington[44].

**Allometry of aerodynamic force**. Allometric equations relate some measurement to (most often) body weight, in the form $Y = aW^b$. We assume that the intraspecific variation we observe is primarily biological, such that we can make meaningful inferences. This implies stable variances on the logarithmic scale, so it is appropriate to log-transform the allometric equation, $\log Y/Y_o = \log a/a_o + b \log W/W_o$. Here, we have preserved the requirement of dimensionless arguments by introducing the characteristic scales $Y_o$, $a_o$ and $W_o$, to obtain reduced dimensions $Y'$, $a'$ and $W'$. The intercept, $\log a/a_o$, is dependent on the choice of characteristic scales. A usual approach is to choose 1 unit of measurement, e.g. 1 g. An alternative reasonable choice is the clade-wide mean of each variable as the characteristic scale for interspecific analyses, and the intraspecific mean for intraspecific analyses. With this choice, the intercept of the linear regression ($\log a'$) must pass through the origin, because the expected values of $\log Y'$ and $\log W'$ are both zero.

The allometric version of the aerodynamic force equation (Eq. (1)) can thus be obtained by equating each term with body weight (omitting the constant of log 1/2). The slopes $b$ are subscripted with the relevant term from the force equation (Eq. (1)), and for simplicity we drop the prime notation.

$$\log_{10}\overline{F}_V = b_{\overline{F}_V} \cdot \log_{10}W = b_\rho \cdot \log_{10}W + 2b_{\overline{U}} \cdot \log_{10}W + b_S \cdot \log_{10}W + b_{\overline{C}_V} \cdot \log_{10}W$$

$$b_{\overline{F}_V} \cdot \log_{10}W = \left(b_\rho + 2b_{\overline{U}} + b_S + b_{\overline{C}_V}\right) \cdot \log_{10}W.$$

Allometric exponents are determined individually, allowing us to take advantage of partly overlapping data sets which may include observations of only some variables. In principle, separation of the problem into components could allow different statistical methods to be applied to each exponent, if warranted[45].

We can infer the statistical validity of the exponents as a group based on whether they correctly predict the relationship of force and body weight, $b_F$ (Supplementary Fig. 1). When the exponents do not sum to $b_F$, some or all of them are likely biased. We cannot provide a hard 'rule' for violation of this constraint, but the magnitude of the difference can help place a minimum bound on the difference from a prediction (e.g. isometry) that can reasonably be considered an allometry. For instance, consider a scenario in which we find that the allometric exponent of wing area versus body mass is 0.57, and that the confidence (or credible) intervals exclude isometry (exponent 0.67). If, however, we also find that the sum of the exponents across the full model of force allometry (Σb) equals 0.90,

then at least one exponent, possibly wing area, is underestimated by a margin that could explain the discrepancy from isometry.

**Induced power calculation**. The mechanical power requirements of flapping flight can be derived using a vortex theory[23] or from a blade element model[46] and are grouped as the aerodynamic (comprising induced and profile power) and inertial components. Profile and inertial powers are strongly dependent on modelling assumptions, and we have therefore focused on induced power, the energy imparted by the bird into its wake.

The induced power can be derived by considering mass flux through the disc area swept out by the wings ($A = \varphi R^2$). Induced power is critical because it is the minimum power required for flight: the muscle must perform work on the wing to add kinetic energy into the slipstream[22]. From conservation of momentum, the induced velocity of the fluid is $\overline{v}_{ind} = \sqrt{(F/2\rho A)} = P^*_{RF}$, the Rankine–Froude specific power estimate (here and elsewhere, $P^* = P/W$). We can express the induced velocity directly as a function of the wing velocity $\overline{U}_{wing}$ through the inflow ratio[22], $\lambda = \overline{v}_{ind}/\overline{U}_{wing}$ which yields the induced power $P^*_{\lambda,ind} = \overline{v}_{ind} = \overline{U}_{wing}$. Assuming constant inflow ratio for hovering flight and like in helicopters and actuator discs in general[22] then $P^*_{\lambda,ind} = \overline{v}_{wing}$. This expression for induced power depends only on the wing velocity, but we can apply Ellington's model to study the possible influence of biologically-relevant morphological and kinematic parameters[16]. Ellington derives temporal ($\tau$) and spatial ($\sigma$) correction factors to the Rankine–Froude induced power, so that $P^*_{ind} = P^*_{RF}(1 + \tau + \sigma)$. The spatial correction factor models how wing morphological variation and kinematics (we assume harmonic motion of the wing) impact the induced wake, and the temporal correction factor models unsteadiness in the wake due to kinematic parameters such as the stroke frequency. Although the correction factors typically alter the induced power estimate by only 10–15%[47] this difference ostensibly could depend on species' and individuals' body masses (perhaps through indirect correlations with morphological variation). The induced power relationships might therefore change in ways that are not expected from the Rankine–Froude estimate alone. Use of $P^*_{\lambda,ind}$ or $P^*_{RF}$ supports our conclusions, though only $P^*_{ind}$ is reported here.

**Interpretation of force coefficient allometry**. Caution is necessary interpreting the slope of $\overline{C}_V$. From dimensional analysis, $\overline{C}_V$ is a scale-free factor, and so cannot depend on body mass over orders of magnitude in size. Within an order of magnitude or less, some progressive changes in $\overline{C}_V$ might contribute to weight support. However, because $\overline{C}_V$ is calculated from other variables, it cannot be distinguished from variable errors on its own, and so if such an effect is present, it must be properly attributed to a cause[18]. Incorporating $\overline{C}_V$ can therefore be viewed, at a minimum, as a check on whether there is a correlation between measurement bias and body weight. However, further detailed studies on the nature of the $\overline{C}_V$ allometry can reveal aspects of the evolution of both wing form and function that are not easily described by the mean dynamic pressure and wing area alone, such as camber or stroke kinematics. Incorporating this term thus serves as a link between readily studied dimensional components and pervasive but less easily quantified functional variation.

Given the computational dependence of $\overline{C}_V$ on the other variables and their errors, it could be argued that the sum-to-one constraint is trivial. This is not the case for this analysis for two reasons. The first is that our exponents are derived from overlapping but not identical data sets. A more general reason is demonstrated through simulations in which we introduce random errors (Gaussian-distributed error with standard deviation equal to 0.1 of the mean) into fixed species means of one or more variables. We then recalculate $\overline{C}_V$ and all exponents, and examine the resulting sum. We do not distinguish between technical and biological error or phylogenetic relatedness, as the emphasis is on any deviation from perfectly predicted exponents. This analysis demonstrates that when only a single variable contains errors, e.g. wing area, the sum-to-one constraint is indeed trivially obeyed (Supplementary Fig. 3, row 1; sum of exponents slightly differs from 1 due to use of empirical data). In this case, the error in $\overline{C}_V$ is simply the error in wing area and so always compensates. When $\overline{C}_V$ absorbs multiple errors, the sum of exponents in any given data set may differ substantially from the true sum, and we find a distribution of possible values (Supplementary Fig. 3, rows 2 and 3).

**Data analysis and mapping**. All analyses were performed with R 3.2.0[48] to organise data and interface with JAGS 4.2[38]). We also used the R package *dplyr*[49] for data manipulation; *ape*, *nlme* and *treespace*[36, 50, 51] for phylogeny manipulation, visualisation of phylogenetic uncertainty, and comparison of our parameter estimates to those obtained by maximum likelihood; and *rjags* and *R2jags*[38, 52] for interfacing with JAGS.

The map in Fig. 1a was generated in R using the packages *mapplots*, *raster*, *rworldmap* and *sp*[53–56]. The map of the Americas, and the latitudes and longitudes of the collections sites, were transformed to a Mollweide projection centred on (Lat 0, Lon −90). For clarity, we omitted collection sites with a single record, and grouped nearby sites (especially transects) in 0.5 × 0.5° cells. The map is shaded to provide elevational context for hummingbird ranges, and the elevation of individual collection sites, relative to 5000 m, is depicted in a cartoon. The phylogeny in Fig. 1b was drawn with the aid of the package *phytools*[57]. The sample size for partial kinematics was the number of individuals with a calculated force

coefficient in hovering, and the sample size for full kinematics was determined as the number of individuals with both a hovering and burst load lifting force coefficient. The sample size for morphology alone was determined as the number of individuals with weight, elevation and wing area data.

**Data availability**. Data reported in this paper and JAGS model specifications are deposited in Figshare database DOI: 10.6084/m9.figshare.5318449.

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

## Acknowledgements

This work was supported by Natural Science and Engineering Research Council of Canada Postgraduate Scholarship and Discovery Grants (RGPIN-2016-05381), the National Science Foundation (IOS-0923849, IOS-1552419, DEB-1146491), and the Air Force Office of Scientific Research (AFOSR-3964930).

## Author contributions

D.A.S. and D.L.A. conceived of the study. D.A.S., D.L., D.L.A., J.W.B. and R.D. developed the modelling and aerodynamic framework. D.L.A., J.A.M., P.S.S., C.C.W., D.G. and K.C.W. collected data. D.A.S. analysed data. D.A.S. and D.L.A. wrote the manuscript. All authors edited the manuscript.

## Additional information

**Competing interests:** The authors have no competing financial interests.

