## [Peer Review File · Nature Communications]

Reviewers' comments:

Reviewer #1 (Remarks to the Author):

Your comprehensive, comparative approach is extraordinary and, overall, it appears that your conclusions about flight performance are consistent with the patterns observed among and within species. Your observation of invariant wing velocity among species is interesting and new. However, overall, I question the novelty of your results for two reasons: since Greenewalt's time, we have understood that hummingbirds exhibit positive allometry for wing area (among species) so this primary conclusion reinforces an already assumed pattern. Also, your empirical evidence for the scaling of maximum force production in load lifting and as a function of altitude has been thoroughly explored (by subsets of your group) in a diverse array of important previous publications such that these aspects of the study do not provide significantly new insight.

In sum, the manuscript reads like a highly useful and laudable review, but not something that fundamentally transforms our understanding of the evolution of avian flight.

Minor Comments: Include in the abstract something about intraspecific trends (e.g. Lines 165-167).

Line 93: "indicating..." this sentence seems to undermine your entire approach. Consider rewording or elaborating with an explanation.

Line 94: specify what is meant by "no appreciable effect"

Line 100: shouldn't "credible" be "confidence"?

Reviewer #2 (Remarks to the Author):

This is an unusual paper. Equation (1) is a definition of the force coefficient, and as such, it does not provide explanation of the measured data. Equation (2) also does not follow equation (1). Given these, the rest of the paper is impossible to interpret.

Reviewer #3 (Remarks to the Author):

This manuscript is a compelling demonstration of the power of biomechanics to inform our understanding of evolution. The authors use a biomechanical model relating morphological and kinematic variables to aerodynamic forces in order to elucidate the scaling of flight mechanics in hummingbirds. A novel aspect of this work is its combination of intraspecific and evolutionary (interspecific) allometry (inferred through phylogenetic comparative methods), and this integration pays off with some intriguing results. Although the scaling of flight forces within species is driven by changes in both wing area and wing velocity, evolutionary allometry seems to be primarily a result of changes in wing area. Looking within species the authors find that increasing wing velocity with size comes at a cost to burst force and energetics. Looking among species, however, the relative stasis of wing velocity suggests that larger species do not suffer these costs. Instead, the flight forces required for larger species seem to be a result of evolutionary increases in wing area. This result is an exciting example of how biomechanics can constrain the path of evolution, and the authors have presented it in a concise and well-written manuscript. Moreover, a series of analyses demonstrate that the result is robust to uncertainty in biomechanical parameters as well as measurement and phylogenetic error.

Although I do not have any major criticisms of this work, I do recommend that the authors consider a

more thorough examination of among clade heterogeneity in allometry. As the authors show in Supplementary Fig. 4, the overall evolutionary allometries do not seem to be driven by any single clade (lines 97-99), but there may still be among-clade differences in these relationships. I was not entirely clear on how clade-specific allometric coefficients were estimated, but if the authors could evaluate standard errors, it should be possible to make some comparisons among clades. Any differences could enrich the story, with some clades exhibiting a different strategy for meeting the demands of changing body size (as suggested on lines 191-192).

Minor comments:

Line 29 Not clear what is meant by "in the absence of further innovations." Further than what?

Line 96 C_w, V has not yet been defined.

Equation 3 Does this require a citation, or are you deriving this relationship here?

Lines 337-338 I like the method for depicting tree uncertainty, but this description is likely to be difficult to follow for a non-specialist. In particular, how do you measure distances between trees and how do you find a median tree?

Fig. 1C It is not clear why some clades of trees i, ii, iii, and iv are highlighted. But a very nice figure otherwise!

Fig. 2 GREAT figure!

Supplementary Fig. 1 I think the caption could more clearly describe what is shown in each of the four panels. It took me some time to determine that the top and bottom panels represent hovering versus burst flight (right?), but I still am not sure what the different colored dots and horizontal bars mean.

We are grateful to the reviewers for sharing their thoughtful critiques of the manuscript, and for the opportunity to respond to the concerns. We have addressed each concern either through additional analysis and/or through edits to the text. The changes are described in point-by-point format here.

REVIEWER 1:

Your comprehensive, comparative approach is extraordinary and, overall, it appears that your conclusions about flight performance are consistent with the patterns observed among and within species. Your observation of invariant wing velocity among species is interesting and new. However, overall, I question the novelty of your results for two reasons: since Greenewalt's time, we have understood that hummingbirds exhibit positive allometry for wing area (among species) so this primary conclusion reinforces an already assumed pattern. Also, your empirical evidence for the scaling of maximum force production in load lifting and as a function of altitude has been thoroughly explored (by subsets of your group) in a diverse array of important previous publications such that these aspects of the study do not provide significantly new insight.

In sum, the manuscript reads like a highly useful and laudable review, but not something that fundamentally transforms our understanding of the evolution of avian flight.

We modified the introduction to better explain the significance of our study, and how the much expanded data set and new analyses address major gaps in our understanding of biomechanical evolution. Central to our work is (1) although some of our previous work has documented divergent performance across elevations, we have lacked a theory to explain how coevolution of many parameters can lead to the patterns we see, and (2) this is the first examination of patterns within species, from which variation among species must ultimately arise.

We have amended the introduction (L45-57) to better explain the ideas. The relevant paragraph now reads,

“Allometries linked to flight performance do not evolve in isolation. The coevolution of suites of biomechanical traits dictates organismal performance, resulting in patterns such as the dependence of flight performance allometry on species elevation¹⁴. The functional evolution of any one trait, such as wing area, must therefore be considered alongside many correlated biomechanical traits. Previous work has especially focused on the evolution of flight performance in response to changes in elevation^{5,14,15}, but a general theory linking this variation to the proximate determinants of flight performance has not yet been developed. Moreover, because allometries are evolving traits, a general understanding of the evolution of flight performance must start at the variation observed among individuals and populations. A barrier to such studies is the daunting number of traits that can potentially be related to flight performance, making it difficult to choose a suite on which to build a complete framework. Simultaneously, the large number of traits might suggest that there are many potential evolutionary paths resulting in similar flight performance. An integrative perspective on this problem must be able to explain not just the presence or absence of an allometry, but also explain its magnitude. We approach this general problem by considering the mechanisms that contribute to the generation and cost of aerodynamic force in flight, and thus develop a framework to unify many aspects of hummingbird flight physiology.”

Include in the abstract something about intraspecific trends (e.g. Lines 165-167).

We have added a statement about the key difference in force allometry within species. The relevant sentence now reads,

“Conversely, wing velocity increases with body weight within species, compensating for lower relative wing area in larger individuals.”

Line 93: “indicating...” this sentence seems to undermine your entire approach. Consider rewording or elaborating with an explanation.

Our intention was to clarify that our method complements but does not supplant decades of research on phylogenetic comparative methods. We have deleted this part of the sentence.

Line 94: specify what is meant by “no appreciable effect”

We made two changes to address this concern. The relevant sentence in the main text now reads,

“Phylogenetic uncertainty, as we model it here, altered mean exponents and credible interval widths by <1%.”

We have also amended Supplementary Figure 2 to numerically quantify the effect of the different models. We expect the modified figure will be of greater impact for the broader community focused on phylogenetic regressions.

Line 100: shouldn’t “credible” be “confidence”?

We have clarified that these are “*Bayesian credible intervals*”, to distinguish from confidence intervals, which are the product of frequentist statistical approaches.

REVIEWER 2:

This is an unusual paper. Equation (1) is a definition of the force coefficient, and as such, it does not provide explanation of the measured data. Equation (2) also does not follow equation (1). Given these, the rest of the paper is impossible to interpret.

We apologise for the confusion. We derive both equations in the supplementary material, but had neglected to point the reader to this important information. We have now amended the text at both equation 1 and equation 2 to point to the model derivation in the Supplementary Methods. We have also updated the text to explain that equation 1 is consistent with, and derived from, the use of the force equation in blade element modelling. The changes are made at the following locations:

The relevant callout in the main text now reads, L64,

“According to a blade element model (developed in Supplementary Methods), the time-averaged equation for vertical, weight-supporting aerodynamic force during hummingbird hovering is,”

The proper interpretation of C_v was not fully developed in the text. We have amended the ‘Supplementary Methods - Interpretation of force coefficient allometry’ to include,

“Incorporating \bar{C}_v can therefore be viewed, at a minimum, as a check on whether there is a correlation between measurement bias and body weight. However, further detailed studies on the nature of the \bar{C}_v allometry can reveal aspects of the evolution of both wing form and function that are not easily described by the mean dynamic pressure and wing area alone, such as camber or stroke kinematics. Incorporating this term thus serves as a link between readily studied dimensional components and pervasive but less easily quantified functional variation.”

We show how equation 2 follows from equation 1 through a derivation presented in the Supplementary Methods. We point the reader to this derivation by adding to L86,

“...where each slope b refers to a variable in equation (1), according to its subscript (derivation presented in Supplementary Methods).”

REVIEWER 3:

This manuscript is a compelling demonstration of the power of biomechanics to inform our understanding of evolution. The authors use a biomechanical model relating morphological and kinematic variables to aerodynamic forces in order to elucidate the scaling of flight mechanics in hummingbirds. A novel aspect of this work is its combination of intraspecific and evolutionary (interspecific) allometry (inferred through phylogenetic comparative methods), and this integration pays off with some intriguing results. Although the scaling of flight forces within species is driven by changes in both wing area and wing velocity, evolutionary allometry seems to be primarily a result of changes in wing area. Looking within species the authors find that increasing wing velocity with size comes at a cost to burst force and energetics. Looking among species, however, the relative stasis of wing velocity suggests that larger species do not suffer these costs. Instead, the flight forces required for larger species seem to be a result of evolutionary increases in wing area. This result is an exciting example of how biomechanics can constrain the path of evolution, and the authors have presented it in a concise and well-written manuscript. Moreover, a series of analyses demonstrate that the result is robust to uncertainty in biomechanical parameters as well as measurement and phylogenetic error.

Although I do not have any major criticisms of this work, I do recommend that the authors consider a more thorough examination of among clade heterogeneity in allometry. As the authors show in Supplementary Fig. 4, the overall evolutionary allometries do not seem to be driven by any single clade (lines 97-99), but there may still be among-clade differences in these relationships. I was not entirely clear on how clade-specific allometric coefficients were estimated, but if the authors could evaluate standard errors, it should be possible to make some comparisons among clades. Any differences could enrich the story, with some clades exhibiting a different strategy for meeting the demands of changing body size (as suggested on lines 191-192).

We thank the reviewer for the suggestion, and have reperformed these analyses to more carefully consider the question. We have better explained our approach to analysing intraclade patterns in the Methods,

“We additionally examined whether there is any evidence that specific clades depart from the overall trends across all hummingbirds. The previous models, allowing for measurement error but not phylogenetic uncertainty, and with the previous uninformative priors, resulted in very wide credible intervals in some clades due to the reduced sample sizes. Because our objective was to find evidence for departures from the overall trend, we therefore used more reasonably informative priors. Following the overall trends, we employed a normal distribution with $\tau=1$ and either a mean of 1, for wing area, or 0, for other variables. Precisions were modelled directly through a weakly informative conjugate gamma prior with shape and scale equal to 10^{-3} .”

With this revised analysis (using informed priors for the intraclade analysis), we find that there is indeed some evidence that the Bee clade of hummingbirds stands apart from the other groups. We have therefore added to the Discussion (L209),

“We therefore applied our method to probe whether there is any evidence of variation in strategies among hummingbird clades. We find that the Bee clade has a uniquely low wing area exponent, comparable to that observed within species (Supplementary Figure 4). This is particularly striking in light of the fact that the Bee clade is the most recently derived and most rapidly diversifying group of hummingbirds²⁰. Combined with the observation that they also have uniquely low variation in body mass, this suggests a hypothesis that physiological diversification in the Bee clade is lagging behind species diversification.”

Line 29 Not clear what is meant by “in the absence of further innovations.” Further than what?

Sorry for the confusion. The edited sentence now reads,
“Hummingbirds sustain hovering, a highly energetically costly behaviour supported by numerous morphological and kinematic innovations.”

Line 96 C_w, V has not yet been defined.

Fixed, thank you.

Equation 3 Does this require a citation, or are you deriving this relationship here?

We have added a citation, thank you.

Lines 337-338 I like the method for depicting tree uncertainty, but this description is likely to be difficult to follow for a non-specialist. In particular, how do you measure distances between trees and how do you find a median tree?

We have revised the Methods to briefly summarise Kendall and Colijn’s method, and how this leads to a convenient way of choosing a summary tree. The relevant section now reads,

“To examine differences among the hypotheses represented in the posterior tree distribution, we describe each tree, T , by Kendall and Colijn’s measure $v\kappa(T) = (1-\kappa)m(T) + \kappa M(T)$, a convex combination of vectors dependent on the tree’s topology and branch lengths, weighted by κ ³⁵. Any two trees are then related by a Euclidean distance $d(T) = \|v\kappa(T1) - v\kappa(T2)\|$. We visualise the broad uncertainty in the phylogenetic hypothesis by projecting the trees’ pairwise distances into two principal coordinates³⁵, clustering of which revealed four subgroups of trees. Assuming each subgroup encapsulates a distinct source of phylogenetic uncertainty, we can summarise this uncertainty by finding the tree that lies at the geometric median of that subgroup, and then comparing this median tree to the species phylogeny of McGuire et al.²⁰ Major topological differences are highlighted in Figure 1c.”

Fig. 1C It is not clear why some clades of trees i, ii, iii, and iv are highlighted. But a very nice figure otherwise!

Thank you. The relevant section of the figure 1 legend now provides this information:

“The majority of uncertainty in species relationships can be represented by two principal coordinates. Individual trees are shown by filled circles and clustered by similarity, and then compared to the Maximum Clade Credibility tree of McGuire et al.²⁰. The major topological differences represented by each cluster are visualised by plotting the median tree corresponding to each cluster (i-iv) and shading the affected clades (following the colour scheme in panel B). Tree variation in this study primarily represents ambiguities in the Hermit and Brilliant clades. We allow for this uncertainty by integrating over many phylogenetic hypotheses.”

Supplementary Fig. 1 I think the caption could more clearly describe what is shown in each of the four panels. It took me some time to determine that the top and bottom panels represent hovering versus burst flight (right?), but I still am not sure what the different colored dots and horizontal bars mean.

We have substantially revised Supplementary Figure 1 for readability and relevance. The revised figure caption now reads,

“Supplementary Figure 1 Comparison of different methods of reconstructing the allometry of force (**A,B**) and specific induced power (**C,D**). **A** In hovering, the force produced is exactly equal to body weight, which we therefore ‘observe’ to be exactly equal to 1 (vertical dashed lines) both among and within species. For predictions derived from equation (2) to be valid for hovering flight, it is necessary that the sum of the posterior distributions of each term (solid lines) match the observed force generation, and must therefore be centered on 1. This condition is met both among (black) and within (red) species. **B** During burst performance, the allometry of force generation may differ from unity. The exponent of the empirically measured burst force (dashed lines) is compared to the reconstructed burst force obtained by summing the exponents of each term as measured during the assay (solid lines). Among and within species, the two methods again substantially agree with each other. **C,D** In this study, the allometry of specific induced power cannot be observed directly, but must be computed either as described by Ellington (1984)¹⁵ and in the Supplementary methods (long-dashed lines), or by summing the contributions of each component in equation (4) (dotted lines). Specific induced power exhibits significant positive allometry in hovering (C) within species, but neither among nor within species during load lifting (D). Distributions smoothed with bandwidth=0.05.”

REVIEWERS' COMMENTS:

Reviewer #1 (Remarks to the Author):

The authors have addressed my concerns and those of the other reviewers. This study advances our understanding of evolution of form and function in a fascinating clade of birds, and it provides a model approach for future studies attempting similar comparative study in other taxa.

Reviewer #3 (Remarks to the Author):

The distinct pattern of allometry within the Bee clade provides a nice illustration of the power of the approach to identify distinct evolutionary patterns. It's an interesting result that fits nicely within the overarching explanation regarding wing area allometry.

The authors have also done a nice job clarifying all points of confusion I raised in my initial review. The one exception is the explanation of the method for visualizing phylogenetic uncertainty (lines 282-291), which provides no definition for κ , m , or M . While I recognize that the authors probably want to avoid a lengthy description of the Kendall and Colijn method, this explanation should either go a little further in its detail or back off from introducing the equations and instead provide a description suitable for a general audience.

REVIEWER #1:

The authors have addressed my concerns and those of the other reviewers. This study advances our understanding of evolution of form and function in a fascinating clade of birds, and it provides a model approach for future studies attempting similar comparative study in other taxa.

Thank you for your constructive review, which has improved our manuscript.

REVIEWER #3:

The distinct pattern of allometry within the Bee clade provides a nice illustration of the power of the approach to identify distinct evolutionary patterns. It's an interesting result that fits nicely within the overarching explanation regarding wing area allometry.

The authors have also done a nice job clarifying all points of confusion I raised in my initial review. The one exception is the explanation of the method for visualizing phylogenetic uncertainty (lines 282-291), which provides no definition for kappa, m, or M. While I recognize that the authors probably want to avoid a lengthy description of the Kendall and Colijn method, this explanation should either go a little further in its detail or back off from introducing the equations and instead provide a description suitable for a general audience.

We appreciate the reviewer's constructive review, and the opportunity to address this final concern. Of the two options proposed, we decided to provide a suitably general description of the procedure. We have revised the relevant section to succinctly describe the more critical aspect of the scoring. This section now reads,

"We examine differences among the hypotheses represented in the posterior tree distribution using the method of Kendall and Colijn^{35,36}. Each tree is encoded by a score that reflects the extent to which the tree is completely described by the lengths or branching pattern of its edges. The set of scores then forms a Euclidean metric space, i.e., the difference between a pair of trees can be found by the difference in their scores."